# Exercise influences the impact of polychlorinated biphenyl exposure on immune function

**Mahesh R. Pillai**[1¤], **K. Todd Keylock**[2], **Howard C. Cromwell**[3], **Lee A. Meserve**[1]*

**1** Dept. of Biological Sciences, Bowling Green State University, Bowling Green, Ohio, United States of America, **2** Dept. of Exercise Science, Bowling Green State University, Bowling Green, Ohio, United States of America, **3** Dept. of Psychology and J.P. Scott Center for Neuroscience, Mind and Behavior, Bowling Green State University, Bowling Green, Ohio, United States of America

¤ Current address: Human Research Protection Program, The University of Toledo, Toledo, Ohio, United States of America

* lmeserv@bgsu.edu

**Data Availability Statement:** The data is available on figshare, DOI: 10.6084/m9.figshare.12768332.

**Funding:** Department of Biological Sciences, Bowling Green State University, Bowling Green, Ohio. The funders had no role in study design, data

## Abstract

Polychlorinated biphenyls (PCBs) are environmental pollutants and endocrine disruptors, harmfully affecting reproductive, endocrine, neurological and immunological systems. This broad influence has implications for processes such as wound healing, which is modulated by the immunological response of the body. Conversely, while PCBs can be linked to diminished wound healing, outside of PCB pollution systems, exercise has been shown to accelerate wound healing. However, the potential for moderate intensity exercise to modulate or offset the harmful effects of a toxin like PCB are yet unknown. A key aim of the present study was to examine how PCB exposure at different doses (0, 100, 500, 1000 ppm i.p.) altered wound healing in exercised versus non-exercised subgroups of mice. We examined PCB effects on immune function in more depth by analyzing the concentrations of cytokines, interleukin-1β (IL-1β), tumor necrosis factor-α (TNF-α), Interleukin-6 (IL-6) and granulocyte macrophage colony stimulating factor (GM-CSF) in these wounds inflicted by punch biopsy. Mice were euthanized at Day 3 or Day 5 after PCB injection (n = 3–6) and skin excised from the wound area was homogenized and analyzed for cytokine content. Results revealed that wound healing was not signficantly impacted by either PCB exposure or exercise, but there were patterns of delays in healing that depended on PCB dose. Changes in cytokines were also observed and depended on PCB dose and exercise experience. For example, IL-1β concentrations in Day 5 mice without PCB administration were 33% less in exercised mice than mice not exercised. However, IL-1β concentrations in Day 3 mice administered 100 ppm were 130% greater in exercised mice than not exercisedmice. Changes in the other measured cytokines varied with mainly depressions at lesser PCB doses and elevations at higher doses. Exercise had diverse effects on cytokine levels, but increased cytokine levels in the two greater doses. Explanations for these diverse effects include the use of young animals with more rapid wound healing rates less affected by toxin exposure, as well as PCB-mediated compensatory effects at specific doses which could actually enhance immune function. Future work should examine these interactions in more detail across a developmental time span. Understanding how manipulating the effects of exposure to

collection and analysis, decision to publish, or preparation of the manuscript.

**Competing interests:** The authors have declared that no competing interests exist.

environemntal contaminants using behavioral modification could be very useful in certain high risk populations or exposed individuals.

## Introduction

Polychlorinated biphenyls (PCB) are environmental contaminants that were manufactured and used in large quantities for over 40 years because of their wide range of applications. However, eventually the harmful effects of these chemicals were observed and their commercial production and use has been prohibited in the US since the 1970s [1]. Aroclor 1254, which was administered to mice in the present study is a mixture of PCB congeners with 54% chlorination and was widely manufactured and released into the environment [2]. Because of their vurtually indestructible nature, PCBs remain in the environment and are known endocrine disruptors [3]. PCB exposure has been shown to cause carcinogenecity, genotoxicity, reproductive toxicity and immunological effects [4–6]. The majority of studies indicate that PCB exposure leads to immunosuppression in mice and humans, however some studies have shown an increase in immunological response [7–9]. Exposure to polychlorinated biphenyls (PCBs) has been shown to have immunotoxic effect and to impair the functioning of several immune responsive cells [10,11]. Previous studies have revealed that PCB can hamper wound healing [12,13]. However, these earlier studies were done using an invertebrate model (e.g., earthworms) with the PCB absorbance through the skin, and the wound sizes were measured only at 24 hours after wound creation. We expanded the present study by examining a well-established mouse model and extending the time duration of the wound healing analysis.

The present study expanded in another novel way by exploring the impact of exercise on immune function in animal subjects exposed to PCB. Researchers are slowly coming to a consensus that physical activity produces a U-shaped response curve, with both no and extreme physical activity having deleterious effects, whereas moderate physical activity has beneficial effects on both overall health and immune system function [14]. Moderate intensity exercise has been shown to provide beneficial effects, retarding development of various chronic conditions like cardiovascular disease, atherosclerosis and chronic inflammation and it also reducing the progression of these diseases and certain neoplasms if they are already present [15,16]. The production of greater concentrations of anti-inflammatory cytokines as a result of exercise and this production has been demonstrated and this is essential for the positive effects of exercise [17]. The effect of exercise has also been studied in older mice in which the immune function is suboptimal resulting in greater susceptibility to and delayed recovery from infection. Older animals also have also been found to have delayed wound healing. Moderate intensity exercise in these animals improved wound healing rates. It also reduced the concentrations of pro-inflammatory cytokines in the wound tissue of these animals [18,19].

The first aim of the present study was to determine whether PCB exposure impairs wound healing and if exercise can reduce some of the negative PCB effects and improve wound healing. To examine the relationship between PCB exposure and exercise in more detail, we measured the concentrations of various cytokines in wound tissue of mice moderately exercised or not exercised, with or without PCB exposure. Various immune responsive cells and the cytokines released by them play an important part in the inflammatory phase of wound healing by not only recruiting other immune cells, but also in the reepithelization process [20]. Previous studies have shown that excessive inflammation with the presence of increased pro-inflammatory cytokines can delay wound healing, especially in aged animals [21,22]; and that

immunologically suppressed mice lacking macrophages and neutrophils experience greater rates of wound healing [23]. Furthermore, it has been shown that exercise improves wound healing rate in obese and aged mice [24]. The concentrations of pro-inflammatory cytokines like TNF-α, KC and MCP-1 have been observed to decrease in aged mice after exercise and these findings correlated with more rapid wound healing rates in these animals. However, this study did not find significant positive effects of exercise on wound healing in younger mice [19]. In an evolutionarily less complex species, PCB administration in earthworms has impaired wound healing and increased allograft rejection [12,13]. Additionally, *in vitro* studies have found PCB to cause enhanced stimulation of neutrophils to produce reactive oxygen species [25]. However, the function of human and mouse macrophages and mouse splenocytes has been shown to be impaired by PCB administration [26–28]. No previous studies have determined the amounts of cytokines in wound tissue of animals administered PCB. The cytokine data from the present study will give us a novel insight into whether exercise can modulate PCB induced modification of cytokine concentrations and thus negating some of the immunotoxic effects of PCB.

## Materials and methods

### Animal care

Female C57BL/6 mice (8 weeks of age) were obtained from Harlan Laboratories, Indianapolis, IN and housed in the Bowling Green State University (BGSU) animal facility. All the animal studies were conducted as approved by the BGSU Institutional Animal Care and Use Committee (IACUC) under protocol no. 10–013. The mice were individually housed in shoe box cages and maintained on a reverse light-dark cycle. They were provided food (Teklad Mouse Breeder Diet 8626, Envigo, Madison, WI) and water ad libitum, and their body weights were recorded daily. After PCB exposure, exercise regimen and wound generation, the animals were frequently monitored for evidence of distress (poor grooming, frantic appearance, poor coat condition, aversion to handling and wound infection). No signs of distress were observed in animals in this study. At the end of testing, animals were euthanized by rapid $CO_2$ inhalation followed by decapitation with a guillotine.

### PCB exposure

All the animals were given 2 weeks to acclimatize to the animal facility before the study began. All the animals in the present study (both not exercised & exercised groups) were administered PCB (Arochlor 1254; Accustandard Inc. New Haven CT, USA) dissolved in corn oil via intraperitoneal injection at a volume of 10 μl/g body weight. PCB doses administered were 0 (PCB 0), 100 (PCB 100), 500 (PCB 500) or 1000 (PCB 1000) μg/g. Previous studies have used similar PCB dosage [29] and intraperitoneal injections [30]. Following acclimatization, the animals in the group not exercised were housed in the animal facility for 3 weeks without exercise and PCB injection was administered 2 days after this 3-week period without exercise was over. Following acclimatization, the animals in the exercised group began moderate exercise for a 3-week period, 2 days after completion of the exercise period they were administered PCB.

### Exercise regimen

The animals not exercised remained in cages without exercise during the 3-week period. The carts holding the cages of these animals were tethered to the table containing the motorized treadmill using large metal clamps, so that these no exercise animals not exercised were exposed to the same sounds and vibrations as the animals in the exercise group. The animals

in the exercised group were exercised by running on a motorized treadmill (Jog-A-Dog model DC6 1H.P.) which was adapted with individual lanes for mice. This running was done for 30 minutes daily, five days a week for 3 weeks and consisted of moderate intensity running at 10–12 m/min with a 6% incline. These exercise regimens were carried out at the beginning of the active period of these animals (0700–0900 as they are on a reverse light-dark cycle). Previous studies have shown that this intensity of exercise effectively alters immune function [19].

## Wound creation and wound size measurement

This procedure was carried out in the animals used in Aim I (Table 1) of the study. Wounds were made two days following PCB exposure. Mice were anesthetized with 2% isoflurane (Iso-flo®) through a cone mask in 100% oxygen at a flow rate of 2–3 L/min during wound creation. Mice were then administered 0.05 µg/g of analgesic Buprenex (Buprenorphine hydrochloride, 0.3 mg/ml, Reckitt Benckiser, Healthcare (UK) Ltd., Hull, England). A similar dose of Buprenex was administered twice daily for 2 days post wounding. A Wahl Peanut Hair Trimmer (Sterling, IL) was used without guides to remove the hair over an area on the upper back, a location inaccessible to the animal, for the creation of wounds. The shaved area was then cleaned with Betadine and 70% ethanol. Wounds were made using a 3.5 mm sterile disposable punch biopsy instrument (Robbins Instruments, Chatham, NJ) to create one full thickness dermal punch resulting in two wounds. A Canon EOS Rebel XTi camera was used to photograph the wound at the same time daily until the wounds were reduced to 10% of original size or for two weeks after the wounds were made, whichever came first. Image J software (version 1.41o, NIH) was used to analyze the wound and compare it to the reference wound, thus allowing comparison between rates of wound healing.

## Euthanasia

The animals were euthanized by rapid $CO_2$ asphyxiation followed by decapitation with a guillotine after the wound size reached 10% of the original wound or two weeks post wounding, whichever came first. A 6.0 mm punch biopsy instrument was used to harvest the wound and the surrounding tissue.

## Wound cytokine analysis

IL-1, IL-6, keratinocyte chemoattractant (KC), monocyte chemoattractant protein-1 (MCP-1), and tumor necrosis factor- α (TNF-α) protein concentrations in wound tissue (Table 2) were determined using a Bio-Plex Pro Mouse Cytokine 23-plex Assay kit (Bio-Rad Laboratories, Inc., Philadelphia, PA).

## Tissue extraction procedure

The wound and the surrounding tissue from animals that were euthanized by $CO_2$ asphyxiation followed by decapitation with a guillotine at Day 3 and Day 5 after wound creation were

**Table 1. Detailed distribution of the mice used for wound healing study.**

|  |  | Number of Animals | |
| --- | --- | --- | --- |
|  | PCB doses (µg/g) | No Exercise | Exercise |
| Wound Healing | 0 | 6 | 6 |
|  | 100 | 6 | 6 |
|  | 500 | 6 | 6 |
|  | 1000 | 6 | 6 |
|  | Total | 24 | 24 |

**Table 2. Detailed distribution of the mice for cytokine analysis study.**

| | Number of Animals | | | |
|---|---|---|---|---|
| | Day 3 | | Day 5 | |
| PCB doses (µg/g) | No Exercise | Exercise | No Exercise | Exercise |
| 0 | 6 | 6 | 6 | 6 |
| 100 | 6 | 6 | 6 | 6 |
| 500 | 6 | 6 | 6 | 6 |
| 1000 | 6 | 6 | 6 | 6 |
| Total | 24 | 24 | 24 | 24 |

harvested using a 6 mm punch biopsy instrument (Robbins Instruments, Chatham, NJ). Previous study has measured cytokine levels in wound tissue at 3 and 5 days after wound creation [19]. The tissue was flash frozen in liquid nitrogen and then stored in -80˚C freezer. This tissue was homogenized using a protocol developed by Frank and Kampfer [31]. The wound tissue was homogenized in an extraction solution containing sterile 1X PBS and an antiprotease buffer, cOmplete, EDTA-free Protease Inhibitor Cocktail (Roche Diagnostics GmbH, Mannheim, Germany). One tablet of cOmplete was added to 50 ml of 1X PBS to prepare the extraction solution. A single wound tissue sample was added to 1 ml of the extraction solution and homogenized using a PowerGen Generator (PowerGen 125, Fisher Scientific, Pittsburg, PA) and a saw tooth blades (7X95mm). The homogenate was centrifuged at 1000 rpm for 10 min at 4˚C. The supernatant thus obtained was aspirated into 3 ml syringes (BD Biosciences, Mexico) and passed through a 1.2 µm Sartorius Minisart Syringe filter (Supelco, Bellfonte, PA). The filtrate (approx. 600 µl) was aliquoted into tubes and stored at -80˚C. A 15 µl aliquot was used for protein assay.

## Protein assays

Protein concentrations of tissue extracts were determined by using Bio-Rad Protein Assay–Dye Reagent Concentrate, following the protocol of Bio-Rad Laboratories, Richmond, CA. Protein assays were done in order to express cytokine concentrations in the samples per milligram of protein. Stock protein standard solution of 1mg/ml was prepared by using bovine albumin purchased from Sigma Chemical Co., St. Louis, MO dissolved in distilled water. Blank and serial dilutions of standard (1 mg/ml, 0.8 mg/ml, 0.6 mg/ml, 0.4 mg/ml and 0.2 mg/ml) were used to generate the standard curve. The 15 µl aliquot of tissue extract was diluted in 30 µl distilled water (total volume 45 µl) and 10 µl of this dilution was used to load a single well of a Costar 96 well plate. All protein assays were done in triplicate. The dye reagent concentrate was diluted 1:4 with distilled water and 200 µl of this diluted dye was added to each of the wells. The plates were incubated on a shaker for 10 min and then absorbance was measured at 595 nm on a Clariostar plate reader (BMG Labtech, GmbH, Ottenberg, Germany) and analyzed using Mars Data Analysis Software (Version 3.01R2, 2013 BMG Labtech). A standard curve was plotted and used for calculating protein concentrations of tissue extracts. The protein concentrations were expressed as mg/ml.

## Cytokine analysis

A Bio-Plex Pro Mouse Cytokine 23-plex Assay kit (Bio-Rad Laboratories, Inc., Philadelphia, PA) was used for measuring cytokine content using a Bio-Plex 200 instrument (lab of Dr. Stanislaw Stepkowski, The University of Toledo, Toledo, OH) and Bio-Plex Manager 6.1 software (Bio-Rad Laboratories, Inc., Philadelphia, PA) was used for machine operation and data

collection. A Bio-Plex handheld magnetic washer (Bio-Rad Laboratories, Inc., Philadelphia, PA) was used for all washing steps. The protocol accompanying the kit was followed. The following cytokines were measured: Eotaxin, G-CSF, GM-CSF, IFN-γ: Interleukins (IL's); IL-1α, IL-1β, IL-2, IL-3, IL-4, IL-5, IL-6, IL-9, IL-10, IL-12(p40), IL-12(p70), IL-13, IL-17A, KC (keratinocyte), MCP-1/MCAF (monocyte chemotactic and activating factor), MIP-1α, MIP-1β (macrophage inflammatory protein), RANTES (regulated on activation, normal T cell expressed and secreted) and TNF-α (tumor necrosis factor alpha).

## Data analysis

Statistical analysis for all data was completed using SPSS software (IBM, 2015, version 23). For the wound size data, general linear model Repeated Measures ANOVA (along with post hoc tests, Bonferroni and Tukey) was performed for PCB, Exercise and Day. Wound size on Day 0 (day of wound creation) over all the groups was normalized to 1 and all the values for the folowing days were sizes compared to Day 0. Error bars are +/- 1 standard error of the mean (SEM). Significance was ascribed to $p \leq 0.05$.

For the cytokine concentration data, general linear model Univariate ANOVA (along with post hoc tests, Bonferroni and Tukey) was performed for PCB, Exercise and Day. Based on the significance of these data, independent T-tests were run between groups within PCB, Exercise and Day. Error bars are +/- 1 standard error of the mean (SEM). Significance was ascribed to $p \leq 0.05$. Note: Before data were selected for running the statistical analysis, in each of the groups the numbers above or below the mean +/- 1 standard deviation (SD) were omitted since there were a few outliers that were drastically skewing the group means.

## Results

### Wound healing: PCB and exercise effects

**Day effects.** Wounds showed signficant size reductions across days. There was a significant main effect of time (F(6) = 223, p<0.001). There were significant differences for the pairwise comparisons from Day 1 to Day 6 of the analysis (see Fig 1) with the wounds decreasing in size in all groups.

**Exercise effects.** Exercise had no signficant impact on wound healing in the control group (0 PCB). However, there was a general pattern in which wound sizes decreased at a greater rate in exercised mice administered no PCB (Fig 2A).

**PCB effects.** Animals that were not exercised did not experience any significant changes or trends in wound healing rates with varying PCB doses. There was a pattern where the rates of wound healing are greater in PCB 1000 administered animals as compared to PCB 0 (Fig 3C), PCB 100 (Fig 3E) and PCB 500 (Fig 3F). Comparison of the other PCB doses did not yield any obvious pattern (Fig 3A, 3B and 3D).

**Exercise and PCB interactions.** Exercise and PCB exposure appeared to interact in diverse ways. In the low dose group exposed to PCB 100 mice (Fig 2B) those not exercised healed at a greater rate than the exercised group. A nonsignificant pattern was revealed where wound sizes decrease at a greater rate in exercised mice administered PCB 500 (Fig 2C). Additionally wound healing rates appear very similar in exercised and not exercised mice administered PCB 1000 (Fig 2D).

No significant changes or trends in wound healing rates with varying PCB doses were observed in animals that were exercised. However, the figures reveal a pattern where the rates of wound healing are less in PCB 100 administered animals as compared to PCB 0 (Fig 4A), PCB 500 (Fig 4D) and PCB 1000 (Fig 4E). No particular pattern was observed with other PCB doses (Fig 4B, 4C and 4F).

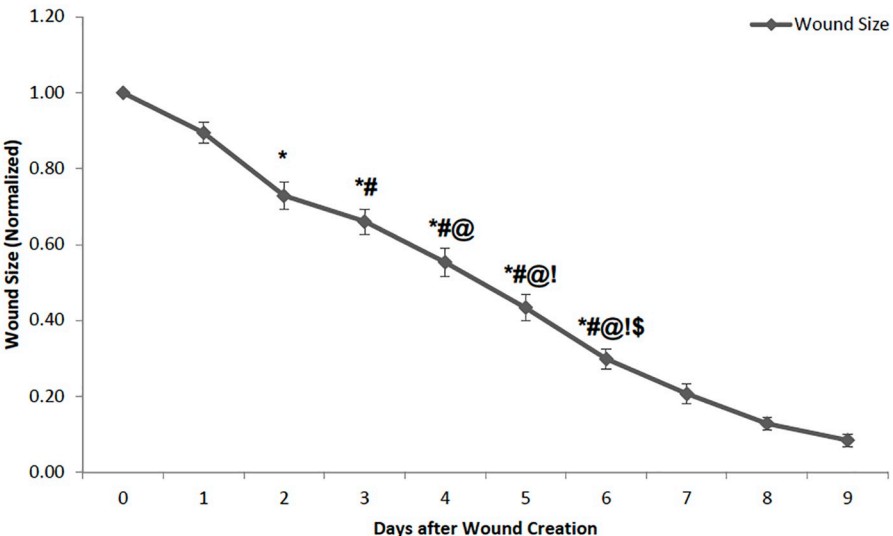

**Fig 1. Comparison of the wound size in animals from day 0 (day of wound creation) to day 9 (n = 11–49).** Animals from all the treatment groups were combined and used to study the day effect. The data have been normalized with the initial wound size on the day of production (Day 0) being 1. As can be expected, the wound size decreased consistently over the period, reduced to less than 10% in 9 days. Wound size is significantly less on all the previous days except for the immediate day before. For example, wound size on Day 3 is significantly less than Day 0 and Day 1, but not Day 2. All the other days followed a similar pattern.

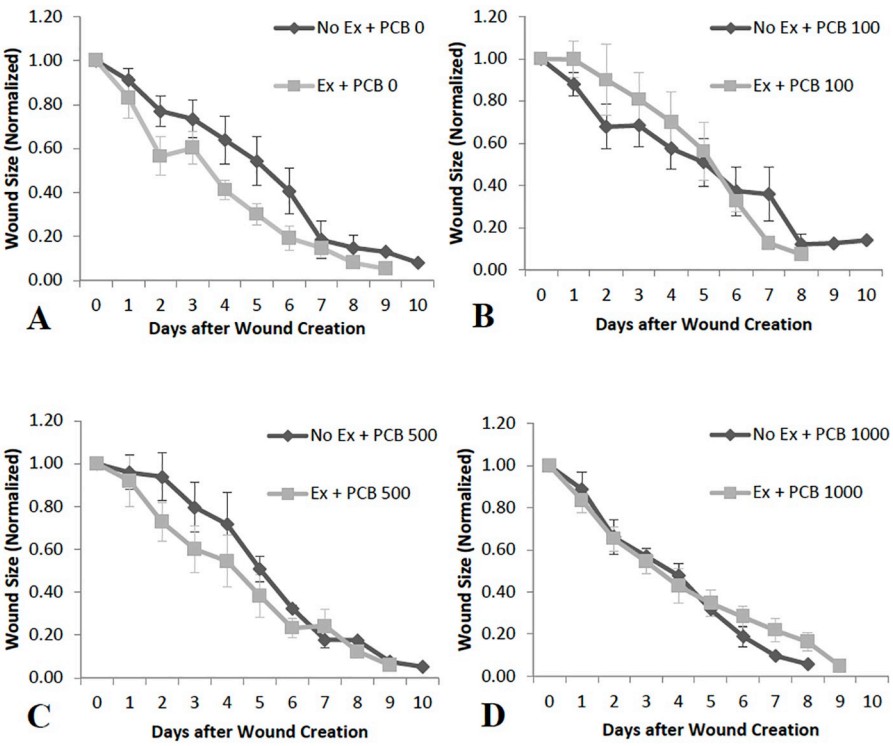

**Fig 2. Effect of exercise on wound size in mice administered varying doses of PCB (n = 4–10).** Animals that were admininstered PCB in varying doses did not show any significant differences or trends in wound size on different days after wound production, irrespective of whether they were exercised or not. However, observation of the graph shows that there seems to be a difference in the mean wound sizes Day 1 through Day 6 between the exercise and no exercise groups with all doses of PCB, except PCB 1000.

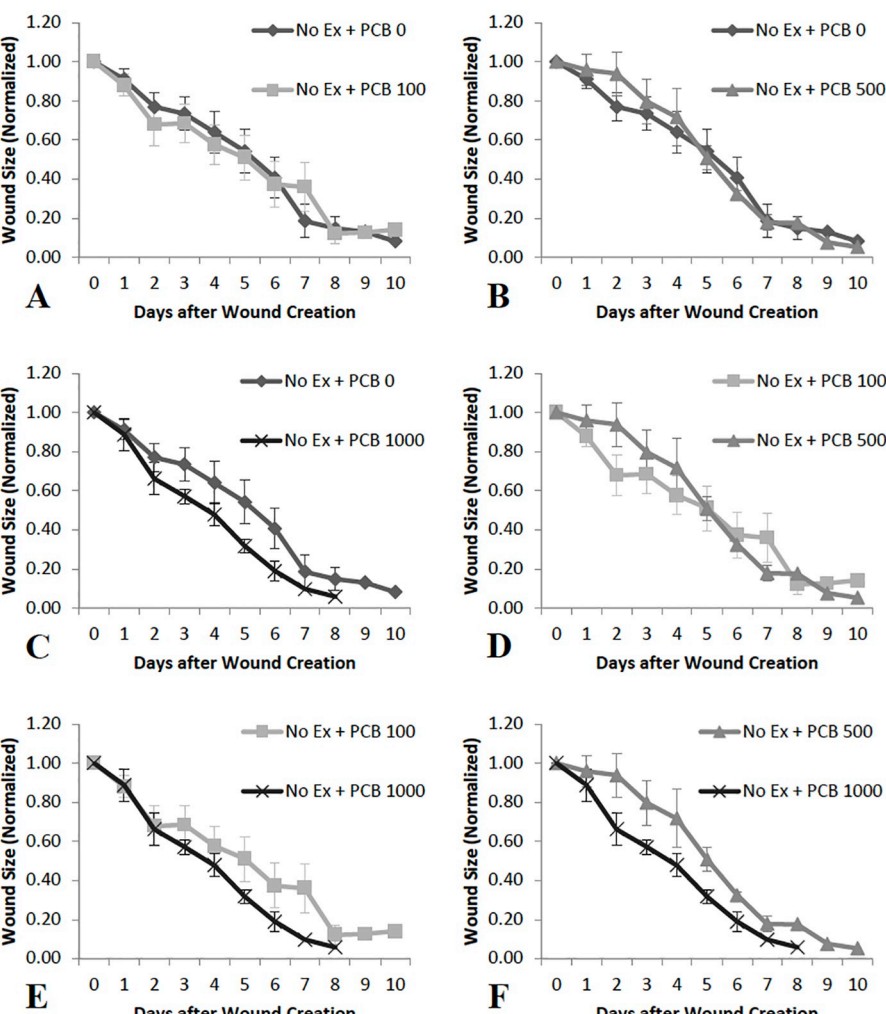

**Fig 3. Comparison of wound sizes on different days in animals not exercised and administered varying doses of PCB (n = 4–10).** C, E and F: The mean wound size is less on Day 2 through Day 6 in PCB 1000 treated animals as compared to PCB 0, PCB 100 and PCB 500 administered mice, though no significant differences or trends were observed. A, B, and D: No significant differences or trends and no difference in means were observed when the other doses of PCB were compared.

## Cytokine concentrations

The cytokine concentrations mentioned in the following sections are concentrations in the wound tissue and not circulating concentrations or whole animal concentrations.

### Effect of exercise and PCB on IL-1β levels

IL-1β concentrations were determined at Day 3 and Day 5 post-wounding (Fig 5). Univariate ANOVA revealed following significant interactions: PCB X exercise ($F_{3,51}$ = 3.956; p = 0.013), PCB X day ($F_{3,51}$ = 3.707; p = 0.017) and exercise X day ($F_{1,51}$ = 8.033; p = 0.007). Independent t-tests were performed to further explore the within group interactions.

**Exercise and day interactions.**   The effects of exercise depended on the day that levels of cytokines were analyzed. Concentrations of IL-1β on Day 5 in mice not administered PCB were significantly less in exercised as compared to not exercised mice (!:$t_5$ = 4.561; p = 0.006).

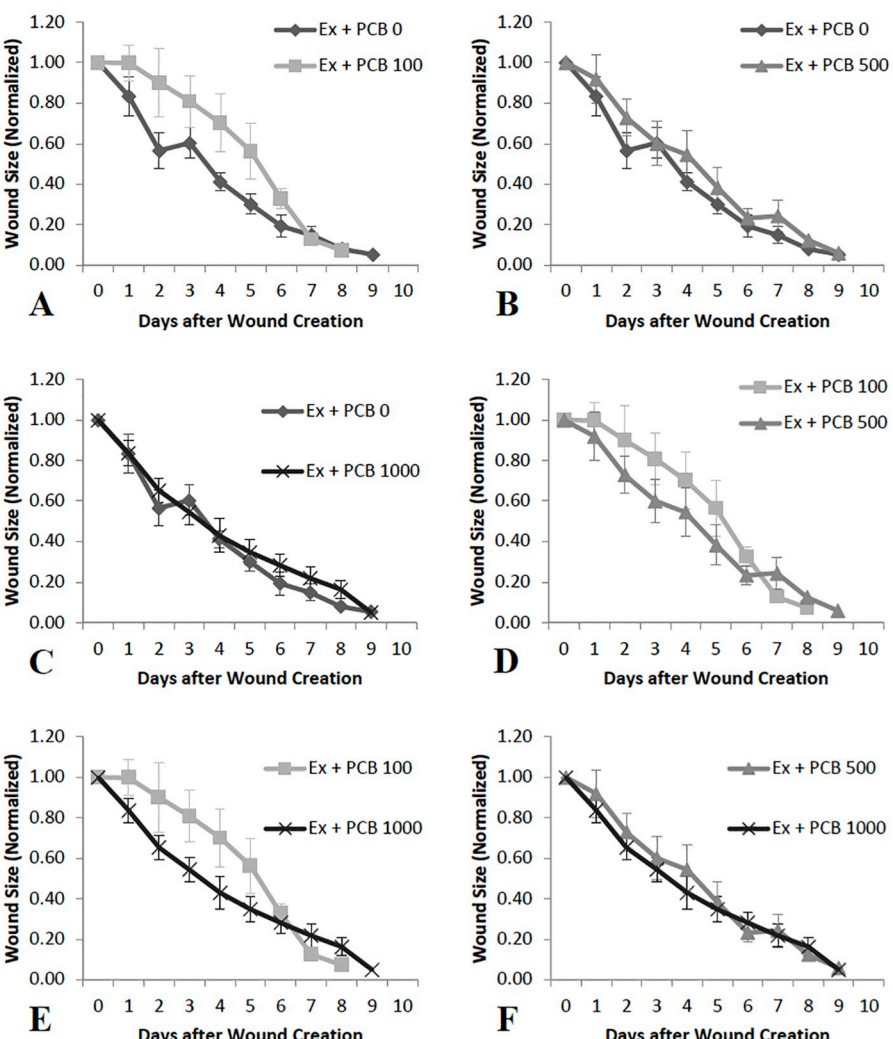

**Fig 4. Comparison of wound sizes on different days in exercised animals administered varying doses of PCB (n = 4–10).** A, D and E: The mean wound size is greater on Day 1 through Day 6 in PCB 100 treated animals as compared to PCB 0, PCB 500 and PCB 1000 administered mice, though no significant differences or trends were observed. B, C, and F: No significant differences or trends and no difference in means were observed when the other doses of PCB were compared.

Exercised mice, not administered PCB, revealed significantly less IL-1β on Day 5 as compared to Day 3 ($^*$:$t_6$ = 3.932; p = 0.008) (Fig 5C and 5D).

**PCB and day interactions.** On Day 3 wound tissue from animals not exercised contained significantly less of IL-1β in mice given 100 μg/g PCB as compared to the animals receiving no PCB (&:$t_6$ = 2.982; p = 0.025) (Fig 5A). In mice not exercised, administered 100 μg/g PCB, significantly greater concentrations of IL-1β were present post-wounding on Day 5 as compared to Day 3 (#:$t_6$ = -2.640; p = 0.039) (Fig 5A and 5B). On post-wounding Day 5, not exercised mice revealed significantly greater concentrations of IL-1β in animals administered 1000 μg/g PCB as compared to mice not administered PCB (@:$t_4$ = -10.193; p = 0.001) (Fig 5B). Mice that were not exercised, on post-wounding Day 5 revealed significantly greater concentrations of IL-1β in 1000 μg/g PCB administered animals as compared to the ones administered 100 μg/g PCB (¤:$t_5$ = -2.898; p = 0.034) (Fig 5B). A between-group comparison showed

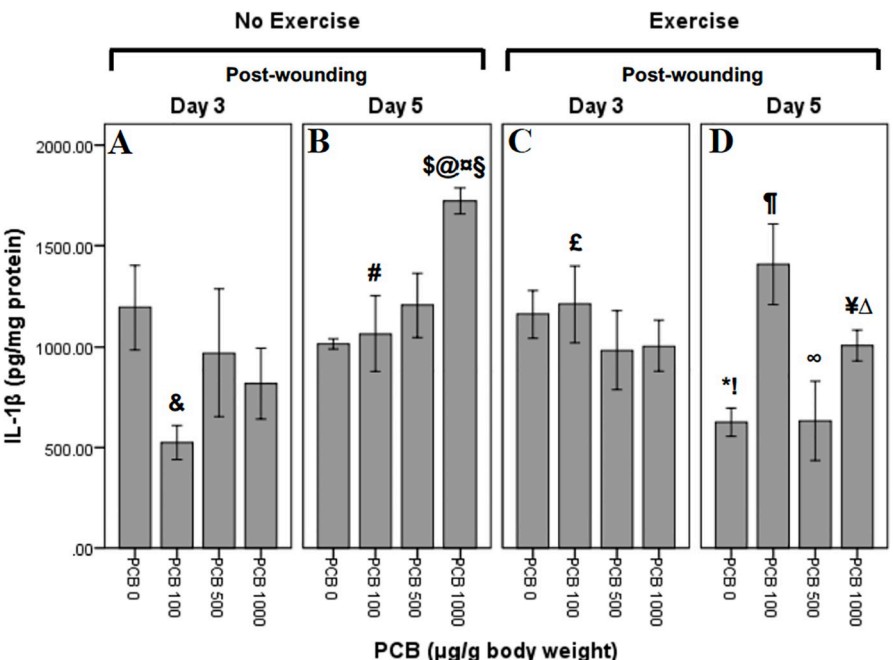

**Fig 5. Effect of exercise and PCB on levels of IL-1β in wound tissue at day 3 and day 5 post-wounding.**
Comparison of IL-1β levels, 3 days post-wounding in animals not exercised reveals a pattern of reduced concentrations with different doses of PCB as compared to no PCB administration, with PCB 100 demonstrating the most reduction. Similar pattern is observed at Day 3 in exercised animals, but with no change in PCB 100 administration. However, on Day 5, IL-1β concentrations in not exercised animals reveals a pattern of dose dependent increase. In exercised animals, on Day 5, concentrations of IL-1β are higher in PCB 100 and 1000 doses, but remain the same with PCB 500 administration.

that post-wounding Day 5 mice that were not exercised revealed significantly greater concentrations of IL-1β in 1000 μg/g PCB treated animals as compared to animals administered 500 μg/g PCB ($:$t_5$ = -2.670; p = 0.044) (Fig 5B). Between group comparison revealed that mice that were not exercised and administered 1000 μg/g PCB, exhibited significantly greater concentrations of IL-1β on post-wounding Day 5 as compared to Day 3 ($:$t_6$ = -3.771; p = 0.009) (Fig 5A and 5B).

**PCB and exercise interactions.** Tissue from mice administered 100 μg/g PCB, contained significantly greater concentrations of IL-1β on Day 3 in exercised animals as compared to animals not exercised (£:$t_6$ = -3.309; p = 0.016) (Fig 5A and 5C). Post-wounding Day 5 animals that were exercised demonstrated significantly lesser concentrations of IL-1β in 500 μg/g PCB administered animals as compared to the ones administered 100 μg/g PCB (∞:$t_7$ = 2.719; p = 0.030) (Fig 5D). On post-wounding Day 5, exercised animals, demonstrated significantly greater concentrations of IL-1β in mice administered 100 μg/g PCB as compared to the animals not administered PCB (¶:$t_6$ = -3.678; p = 0.010) (Fig 5D). On Day 5, mice administered 500 μg/g PCB, displayed a trend towards less IL-1β in exercised as compared to not exercised animals ($t_7$ = 2.173; p = 0.066) (Fig 5B and 5D). Concentrations of IL-1β on Day 5 in mice administered 1000 μg/g PCB were significantly less in exercised mice as compared to mice not exercised (¥:$t_5$ = 6.722; p = 0.001) (Fig 5B and 5D). Mice that were exercised, on Day 5 post-wounding revealed significantly greater concentrations of IL-1β in 1000 μg/g PCB administered animals as compared to the ones not administered PCB (Δ:$t_6$ = -3.659; p = 0.011) (Fig 5D).

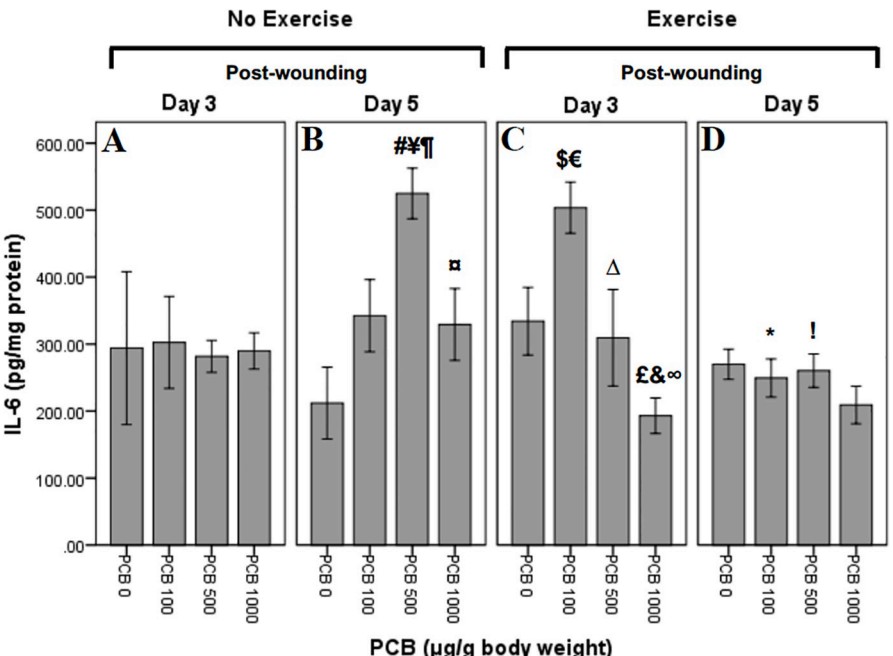

**Fig 6. Effect of exercise and PCB on levels of IL-6 in wound tissue at day 3 and day 5 post-wounding.** In not exercised animals, on Day 3, there appears to be no difference in the means of IL-6 concentrations across PCB doses. However, Day 3, exercised animals demonstrate a pattern of reduced IL-6 concentrations with PCB 500 and PCB 1000, but increased with PCB 100 administration as compared to PCB 0. On Day 5, in not exercised mice, IL-6 concentrations appear to be greater with all PCB doses when compared to PCB 0, with PCB 100 being the highest. Whereas, Day 5, exercised animals reveal pattern of slight decrease with all PCB doses as compared to no PCB administration.

## Effect of exercise and PCB on IL-6 levels

IL-6 levels were measured at Day 3 and Day 5 post-wounding (Fig 6). Univariate ANOVA revealed significant PCB effect ($F_{3,56}$ = 3.008; p = 0.038). It also revealed significant interactions of the following: PCB X exercise ($F_{3,56}$ = 3.113; p = 0.033), PCB X day ($F_{3,56}$ = 3.103; p = 0.034) and exercise X day ($F_{1,56}$ = 7.400; p = 0.009). There was a trend towards PCB X exercise X day interaction ($F_{3,56}$ = 2.408; p = 0.077). Independent t-tests were performed to further explore the within group interactions.

**Exercise and day interactions.** There were no significant exercise and day interactions.

**PCB and day interactions.** On post-wounding Day 5, mice not exercised revealed significantly greater concentrations of IL-6 in animals administered 500 μg/g PCB as compared to mice not administered PCB (¥:t7 = -4.922; p = 0.002) (Fig 6B). Post-wounding Day 5 animals that were not exercised revealed significantly greater concentrations of IL-6 in 500 μg/g PCB administered animals as compared to the ones administered 100 μg/g PCB (¶:t7 = -2.862; p = 0.024) (Fig 6B). Mice that were not exercised, on post-wounding Day 5 demonstrated lower concentrations of IL-6 in 1000 μG/G PCB administered animals as compared to 500 μg/g PCB administered mice (¤:t7 = 3.073; p = 0.018) (Fig 6B). In mice that were not exercised and administered 500 μg/g PCB, there were significantly greater concentrations of IL-6 on post-wounding Day 5 as compared to Day 3 (#:$t_6$ = -5.447; p = 0.001) (Fig 6A and 6B).

**PCB and exercise interactions.** Exercised animals, on Day 3 post-wounding, revealed significantly greater concentrations of IL-6 in mice administered 100 μg/g PCB as compared to the animals not administered PCB (€:t7 = -2.737; p = 0.029) (Fig 6C). Mice that were exercised,

on Day 3 post-wounding demonstrated significantly lesser concentrations of IL-6 in 1000 μg/g PCB administered animals as compared to the ones not administered PCB (&:t8 = 2.722; p = 0.026) (Fig 6C). Mice that were exercised, on Day 3 post-wounding revealed significantly lesser concentrations of IL-6 in 500 μg/g PCB administered animals as compared to 100 μg/g PCB administered mice (Δ:t8 = 2.382; p = 0.044) (Fig 6C). Post-wounding Day 3 mice that were exercised revealed significantly lesser concentrations of IL-6 in 1000 μg/g PCB treated animals as compared to animals administered 100 μg/g PCB ($\infty$:t9 = 6.883; p < 0.001) (Fig 6C). Concentrations of IL-6 on Day 3 post-wounding in mice administered 100 μg/g PCB were significantly greater in exercised mice as compared to mice not exercised ($:$t_8$ = -2.564; p = 0.033) (Fig 6A and 6C). In mice that were exercised and administered 100 μg/g PCB, there were significantly lesser concentrations of IL-6 on Day 5 post-wounding as compared to Day 3 (*:t8 = 5.344; p = 0.001) (Fig 6C and 6D). In mice administered 500 μg/g PCB, on post-wounding Day 5 revealed significantly lower concentrations of IL-6 in exercised animals as compared to animals not exercised (!:$t_7$ = 5.502; p = 0.001) (Fig 6B and 6D). On Day 3 post-wounding, mice administered 1000 μg/g PCB, revealed significantly lower concentrations of IL-6 in exercised animals as compared to animals not exercised (£:$t_8$ = 2.458; p = 0.039) (Fig 6A and 6C).

## Effect of exercise and PCB on KC levels

KC levels were measured at Day 3 and Day 5 post-wounding (Fig 7). Univariate ANOVA did not reveal any significant individual or group interactions. Independent t-tests were performed to explore any within group interactions.

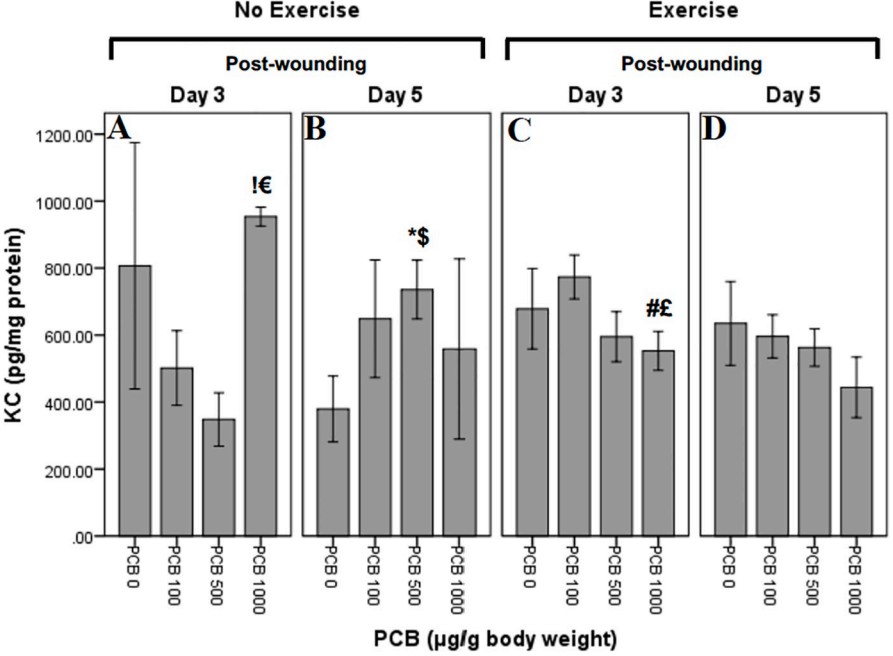

**Fig 7. Effect of exercise and PCB on levels of KC in wound tissue at day 3 and day 5 post-wounding.** KC concentrations on Day 3 in not exercised animals reveal pattern of reduction with increasing PCB doses, except PCB 1000 where there is an increase. Similar pattern of reduction in KC concentrations with increasing dose is seen on Day 3, exercised animals, except for PCB 100 which demonstrates an increase. On Day 5, in not exercised animals, there is a pattern of greater KC with all PCB does as compared to PCB 0, with PCB 500 revealing the highest concentrations. However, exercised animals on Day 5 demonstrate a pattern of dose dependent reduction in KC concentrations.

**Exercise and day interactions.** There were no significant exercise and day interactions.

**PCB and day interactions.** Animals not exercised, on Day 5 post-wounding, demonstrated significantly greater concentrations of KC in mice administered 500 μg/g PCB as compared to the animals not administered PCB ($\$:t5 = -2.585$; $p = 0.049$) (Fig 7B). Post-wounding Day 3 animals that were not exercised revealed considerably lesser concentrations of KC in 1000 μg/g PCB administered animals as compared to the ones administered 500 μg/g PCB (€:$t_7 = -5.122$; $p = 0.001$) (Fig 7A). In mice that were not exercised and administered 500 μg/g PCB, there were significantly greater concentrations of KC on Day 5 post-wounding as compared to Day 3 (*:$t_7 = -2.978$; $p = 0.021$) (Fig 7A and 7B). On post-wounding Day 3, mice not exercised revealed significantly greater concentrations of KC in animals administered 1000 μg/g PCB as compared to mice administered 100 μg/g PCB (!:$t_6 = -3.014$; $p = 0.024$) (Fig 7A).

**PCB and exercise interactions.** Mice that were exercised, on Day 3 post-wounding revealed considerably lesser concentrations of KC in 1000 μg/g PCB administered animals as compared to the ones administered 100 μg/g PCB (£:$t_7 = 2.520$; $p = 0.040$) (Fig 7C). On Day 3 post-wounding, mice administered 100 μg/g PCB, revealed trend towards more KC in exercised animals as compared to animals not exercised ($t_7 = -1.956$; $p = 0.091$) (Fig 7A and 7C). In mice administered 500 μg/g PCB, on post-wounding Day 3 demonstrated trend towards more KC in exercised animals as compared to animals not exercised ($t_9 = -2.222$; $p = 0.053$) (Fig 7A and 7C). Concentrations of KC on Day 3 post-wounding in mice administered 1000 μg/g PCB were considerably less in exercised mice as compared to mice not exercised (#:$t_6 = 5.011$; $p = 0.002$) (Fig 7A and 7C).

## Effect of exercise and PCB on MCP-1 levels

MCP-1 levels were measured at Day 3 and Day 5 post-wounding (Fig 8). Univariate ANOVA revealed significant day effect ($F_{1,53} = 8.623$; $p = 0.005$). It also revealed significant PCB X day interaction ($F_{3,53} = 3.753$; $p = 0.016$) and significant exercise X day interaction ($F_{1,53} = 4.247$; $p = 0.044$). Independent t-tests were performed to further explore the within group interactions.

**Exercise and day interactions.** There were no significant exercise and day interactions.

**PCB and day interactions.** Animals not exercised, on Day 5 post-wounding, revealed significantly greater concentrations of MCP-1 in mice administered 100 μg/g PCB as compared to the animals not administered PCB ($\$:t6 = -3.320$; $p = 0.016$) (Fig 8B). On post-wounding Day 5, mice not exercised revealed trend towards more MCP-1 in animals administered 500 μg/g PCB as compared to mice not administered PCB ($t7 = -2.279$; $p = 0.057$) (Fig 8B). Mice not exercised, on Day 3 post-wounding demonstrated significantly greater concentrations of MCP-1 in 1000 μg/g PCB administered animals as compared to the ones administered 500 μg/g PCB (!:$t7 = -2.448$; $p = 0.044$) (Fig 8A). In mice that were not exercised and administered 500 μg/g PCB, there was a trend towards more MCP-1 on Day 5 post-wounding as compared to Day 3 ($t_8 = -2.299$; $p = 0.051$) (Fig 8A and 8B).

**PCB and exercise interactions.** Day 5 post-wounding animals that were exercised revealed trend towards less MCP-1 in 1000 μG/G PCB administered animals as compared to the ones administered 500 μg/g PCB ($t_5 = 2.532$; $p = 0.052$) (Fig 8D). Concentrations of MCP-1 on post-wounding Day 3 in mice administered 500 μg/g PCB were significantly greater in exercised as compared to not exercised mice (#:$t_7 = -2.460$; $p = 0.043$) (Fig 8A and 8C). In mice that were exercised and administered 1000 μg/g PCB, there were considerably lesser concentrations of MCP-1 on Day 5 post-wounding as compared to Day 3 (*:$t_8 = 3.088$; $p = 0.015$) (Fig 8C and 8D).

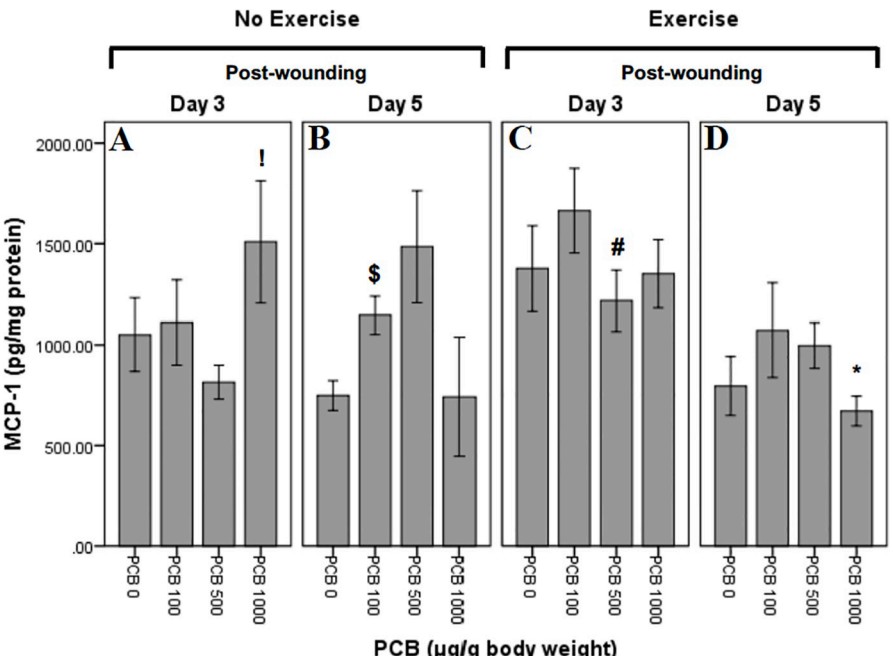

**Fig 8. Effect of exercise and PCB on levels of MCP-1 in wound tissue at day 3 and day 5 post-wounding.** On Day 3, not exercised animals demonstrate a pattern of increased MCP-1 concentrations with PCB administration as compared to no PCB administration, except PCB 500 which reveals a decrease. However, on Day 3, in exercised animals, a pattern of reduced MCP-1 concentrations with PCB administration is observed, except PCB 100 where it increases. Day 5, not exercised animals demonstrate a pattern of dose dependent increase in MCP-1 concentrations, except PCB 1000 which is similar to PCB 0. Similarly, exercised animals, on Day 5, have greater MCP-1 concentrations with PCB administration, except PCB 1000 which is lower.

### Effect of exercise and PCB on TNF-α levels

TNF-α levels were measured at Day 3 and Day 5 post-wounding (Fig 9). Univariate ANOVA did not reveal any significant individual or group interaction effects. Independent t-tests were performed to further explore the within group interactions.

**Exercise and day interactions.** There were no significant exercise and day interactions.

**PCB and day interactions.** Not exercised animals, on post-wounding Day 3, demonstrated considerably lesser concentrations of TNF-α in mice administered 100 µg/g PCB as compared to the animals not administered PCB (*:t6 = 2.838; p = 0.030) (Fig 9A).

**PCB and exercise interactions.** On Day 3 post-wounding, exercised mice revealed trend towards less TNF-α in animals administered 100 µg/g PCB as compared to mice not administered PCB ($t_8$ = 2.187; p = 0.060) (Fig 9C). Mice that were exercised, on Day 3 post-wounding revealed considerably lesser concentrations of TNF-α in 500 µg/g PCB administered animals as compared to the ones not administered PCB (#:$t_9$ = 2.449; p = 0.037) (Fig 9C). Post-wounding Day 3 animals that were exercised revealed trend towards less TNF-α in 1000 µg/g PCB administered animals as compared to the ones not administered PCB ($t_9$ = 2.169; p = 0.058) (Fig 9C).

## Discussion

There are many factors that can influence wound healing and related immune responses. These include the diverse factors of 1) time period following injury, 2) age when injury occurred, 3) sex of the subject, 4) hormonal status (in the present work this could be shifted

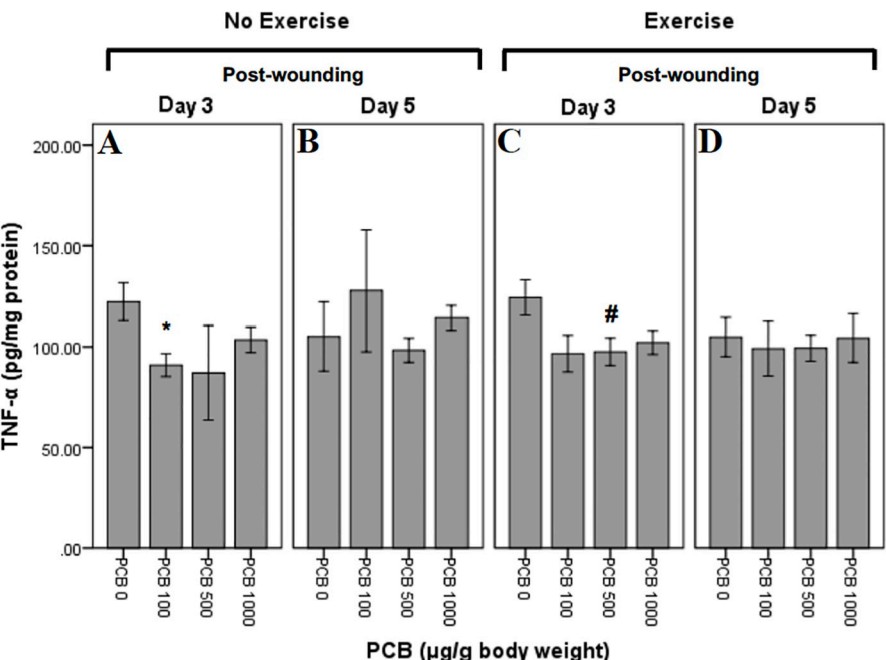

**Fig 9. Effect of exercise and PCB on levels of TNF-α in wound tissue at day 3 and day 5 post-wounding.** On Day 3, not exercised animals demonstrate reduced TNF-α concentrations with varying doses of PCB administration as compared to no PCB. Similarly, reduced TNF-α concentrations were seen with PCB administration in exercised animals on Day 3. On Day 5, in not exercised animals, PCB 100 revealed a slight increase in TNF-α concentrations, whereas PCB 500 and PCB 1000 demonstrated little or no change when compared to no PCB. However, on Day 5, exercised mice, reveal little or no change with varying doses of PCB.

due to exposure of a known endocrine disruptor), and most importantly, 5) the intensity and duration of different components or stages of immunoregulation. The extent of research on these different influences is vast but much more work needs to be completed in order to understand how these factors interact with specific environmental experiences such as exposure to toxicants or exercise. Despite endocrine disruption being a pervasive experience with many diverse compounds in the environment [32] little is known about the relationship between wound healing and immune responses following exposure to these compounds. Similarly, exercise effects have been well documented yet interactions with exposure to endocrine disruption has been widely neglected. We will discuss each of these factors in turn as they interact with either exercise or exposure to PCB as well as discuss our findings for how exercise and PCB combine to alter wound healing and the immune reaction that follows.

There is an abundant amount of data showing that wound healing depends upon time-dependent expression of a variety of physiological signals and processes [33]. One of the complicating factors involved in studying diverse influences on healing is the different time scales that factors primarily operate. Exercise effects are time-dependent and can have several effects relatively rapidly, including an impact on immune function, during exercise and in the hours following [34]. Several of these effects are very short-lived and other effects that are more gradual in nature are longer lasting [35]. In comparison, exposure to endocrine disruptors such as PCBs lead to slower effects that can persist highly dependent upon the concentration of the toxin exposure [36,37]. Our doses were chosen based on this previous work showing that exposure to a similar concentration range of PCB alters hormone function and possibly could have an influence on immune responses. A major difference between the present work and the

majority of the previous studies examining PCB effects was the age of the animal subject being studied.

Quite a bit is known about interactions between age and environmental factors of endocrine disruption per se [38–40]. Many studies have found that earlier exposure in the perinatal period leads to the greatest alterations in behavior and physiology [41,42]. This early period is characterized with vulnerable physiological and anatomical development including brain regions involved in cognition, emotion and motivation. Exercise effects have also been shown to be age-dependent [43,44]. Several of the effects are more pronounced in the aged animal compared to the young, supporting the idea that exercise benefits include slowing of certain aging processes. Immune function seems to also fit into this realm. Wound healing rates have been shown to be faster in aged mice that were exercised as compared to sedentary mice [19]. As observed in this previous study, wound healing data from the present study, which also used young adult animals, revealed a pattern of faster wound healing in exercised animals as compared to the mice not exercised. This pattern was observed in mice either dosed with no PCB and PCB at 500 μg/g. PCB administered at 100 μg/g resulted in a reversed pattern of slower wound healing in exercised mice. The effects may have been even more significant if exercise was completed at an older age. Combining these two factors of PCB and exercise was especially diffcult because of the need to study the younger animal subjects in order to see the most potent PCB-induced effects. Future work could explore earlier toxin exposure with a testing regimen that is expanded across the lifespan of the subject.

Sexual dimorphism of toxic effects of certain endocrine disruptors remains an important question and topic of study [45]. There is evidence that shows male subjects are more vulnerable and show significant changes in reproductive function and behavior [45,46]. When the animals were exposed to PCB in early development (e.g. neonatal nursing period) sex differences are clearly obtained that include sex-specific upregulation of several inflammatory signaling compounds [47]. Females exposure to PCB had significantly higher signaling molecules such as chemokine (C-C motif) ligand 22 and increased gene expression from hypothalamic tissue. Males also had increases in certain components of the immune response making the two profiles similar but distinct. Our study focuses on females and this previous work shows that these animals do show altered immune responses that could impact wound healing and other immune reactions.

Sex differences have also been found in work examining exercise effects [48]. Females have been shown to benefit more from exercise experiences compared to males [49]. These previous studies used different methods and examined different age periods compared to the present work yet support the use of females due to the significant alterations found in these animals that point to harmful effects of toxin exposure and the possible benefits of exercise [50].

PCBs are potent endocrine disruptor compounds (EDCs). Recently key characteristics of EDCs have been determined focusing on mechanisms of EDCs that alter hormone synthesis, transport, receptor signaling and endocrine related epigenetic modifications [51]. Our previous work has focused on PCBs alteration of thyroid function [36,37,52]. There is support for PCB exposure leading to a hypothyroid state, one during development that can lead to impairment of organ function [53]. PCBs have also been shown to alter other hormones including corticosterone and testosterone function [5,54]. Exercise, on the other hand, has been shown to boost or enhance the functioning of these same endocrine functions [55]. In particular, exercise can boost thyroid function and act to ameliorate thyroid deficits including hypothyroid states [56,57]. This present study provides a foundation for future work to examine these types of relationships.

Our findings focused on wound healing and immune function as endpoints to measure PCB and exercise, but other endpoints such as hormone status could be examined in order to

reveal possible benefits of exercise outside of immune responses. Relevant to the present results, thyroid status does have an impact on wound healing and related immune responses [56]. Based on this, PCB as a hypothyroid EDC should have led to a significant change in healing. The other factors including the age and sex of the animals reduced the impact of the PCB and lead a set of diverse interactions between PCB, exercise and the immune response.

Macrophages are critical to many processes, including wound healing. An inflammatory state can increase numbers of macrophages and alter their function. In some populations, moderate-intensity exercise reduced levels of MCP-1 in the wound, which may have led to decreased number of macrophages in the wound [19]. Additionally, PCBs have been demonstrated to impair the survival and function of macrophages [26]. There is research evidence that absence or reduced concentration of macrophages increases rate of wound healing [23]. It is possible that this effect on macrophages is only seen at greater doses of PCBs, such as 1000 μg/g in the present study. A macrophage-mediated effect could also bring about the result in the present study that PCB 1000 μg/g caused a pattern of most rapid wound healing in animals not exercised as compared to all other PCB doses. In the exercised animals, wound healing occurred less rapidly in the animals given PCB at 100 μg/g as compared to the animals administered other PCB doses.

Cytokines play an important role in the inflammatory phase of wound healing [58,59]. It has been previously demonstrated that decreases in pro-inflammatory cytokines in the wound tissue can accelerate wound healing [19,60]. In this study, increased inflammatory cytokines on Day 3 in exercised mice administered 100 μg/g PCB, may be the reason for depressed wound healing rates observed initially as compared to the mice that were not exercised (Fig 2B). Also, it may be that the decrease seen in inflammatory cytokines on Day 5 in PCB administered animals that were not exercised is the reason that wounds healed more rapidly in animals administered 1000 μg/g PCB as compared to the other PCB doses (Fig 3C, 3E and 3F). Day 5 post-wounding in mice not exercised and Day 3 post-wounding in exercised mice demonstrated a similar pattern of IL-6, KC and MCP-1 cytokine content, which could be the result of the inflammation phase occurring at an earlier stage in exercised mice, contributing to the pattern of more rapid wound healing observed in exercised animals administered 0 μg/g PCB, 500 μg/g PCB and 1000 μg/g PCB (Fig 2A, 2C and 2D).

Previous studies have also suggested that reducing the strength of response and length of the inflammatory phase or hastening the resolution phase of wound healing could lead to more rapid wound healing [23,61]. In this study, exercise had diverse effects on cytokine levels, but increased cytokine levels in the two greater doses. Explanations for these diverse effects include the use of young animals with more rapid wound healing rates less affected by toxin exposure, as well as PCB-mediated compensatory effects at specific doses which could actually enhance immune function.

The findings demonstrate the complex nature of interactions that can occur following PCB exposure. The effects of PCB can vary depending upon the dose and duration of exposure. Many of the effects of PCBs and other EDCs are non-monotonic and our study supports this dose-response relationship. The rationale for these nonlinear functions are diverse. One perspective, hormesis, emphasizes the role of compensatory mechanisms occurring at different doses and not others [62]. The ideas of hormesis in toxicology emphasize that exposure to low doses of toxicants leads to an adaptive response that protects or buffers exposures to higher doses [63,64]. The adaptive nature of this low dose is controversial but recent work has supported the idea of a hermetic zone for certain synthetic chemical toxicants [65]. The present findings only partially fit this idea of hormesis. Some aspects of the immune reaction were enhanced at lower doses while the same responses were unchanged or altered in the opposite direction at higher doses. These types of relationships are becoming more common in work

on toxicology and endocrine disruption. The present study reveals possibilities for applying unique methods to reduce harmful effects of environmental toxins. There has been very little work on this idea yet a few studies have found exercise to positively influence exposure to environmental compounds [66,67]. Taking these novel approaches to tackle harmful effects of environmental toxins is required and the methods and findings will certainly be complex yet move the field forward to understanding how to implement new strategies most effectively.

## Acknowledgments

Thanks to Dr. Pavel Anzenbacher Department of Chemistry and his student Elena for use of plate reader for protein assay; Dr. Stanislaw Stepkowski, Medical Microbiology and Immunology, The University of Toledo and his student Dulat for use of Bio-Plex; Vipaporn Phuntumart, Department of Biological Sciences, Bowling Green State University for use of lab space and resources; Dr. Gary Ross, Bio-Rad provided guidance to run cytokine assays.

## Author Contributions

**Conceptualization:** Mahesh R. Pillai, K. Todd Keylock, Howard C. Cromwell, Lee A. Meserve.

**Data curation:** Mahesh R. Pillai.

**Formal analysis:** Mahesh R. Pillai, Howard C. Cromwell.

**Funding acquisition:** Mahesh R. Pillai, Lee A. Meserve.

**Investigation:** Mahesh R. Pillai.

**Methodology:** Mahesh R. Pillai, K. Todd Keylock, Lee A. Meserve.

**Project administration:** Mahesh R. Pillai, K. Todd Keylock, Lee A. Meserve.

**Resources:** K. Todd Keylock, Lee A. Meserve.

**Software:** Howard C. Cromwell.

**Supervision:** K. Todd Keylock, Howard C. Cromwell, Lee A. Meserve.

**Validation:** Mahesh R. Pillai, Howard C. Cromwell, Lee A. Meserve.

**Writing – original draft:** Mahesh R. Pillai.

**Writing – review & editing:** Mahesh R. Pillai, K. Todd Keylock, Howard C. Cromwell, Lee A. Meserve.

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
