## [Decision Letter · Decision Letter 0]

7 May 2020

PONE-D-20-09774

Exercise influences the impact of polychlorinated biphenyl exposure on immune function

PLOS ONE

Dear Dr. Pillai,

Thank you for submitting your manuscript to PLOS ONE. After careful consideration, we feel that it has merit but does not fully meet PLOS ONE’s publication criteria as it currently stands. Therefore, we invite you to submit a revised version of the manuscript that addresses the points raised during the review process.

Both reviewers felt that the presentation of the data is confusing, that the discussion presents mostly results, and that the conclusions are not immediately evident.  In particular, the description of non-significant outcomes could be omitted to make key results from the study more accessible to the readers. Overall, the manuscript needs to be revised to present the results more cohesively. Such a revision will also avoid an overinterpretation of findings that are statistically significant but biologically not relevant. 

We would appreciate receiving your revised manuscript by Jun 21 2020 11:59PM. To enhance the reproducibility of your results, we recommend that if applicable you deposit your laboratory protocols in protocols.io, where a protocol can be assigned its own identifier (DOI) such that it can be cited independently in the future. For instructions see: http://journals.plos.org/plosone/s/submission-guidelines#loc-laboratory-protocols

We look forward to receiving your revised manuscript.

Kind regards,

Hans-Joachim Lehmler, PhD

Academic Editor

PLOS ONE

2. We noted that one of your references did not auto populate and instead the manuscript contains the following  "Citation", please replace this with the appropriate reference during your next revision.

3. At this time, we request that you  please report additional details in your Methods section regarding animal care, as per our editorial guidelines:

(1) Please state the number of mice used in the study  

(2) Please include the secondary and confirmatory method of euthanasia, in addition to CO2 asphyxiation

(3) Please provide the dosage of isoflurane used to anaesthetise the mice during the wound creation experiment

(3) Please describe the post-operative care received by the animals in both the wound creation and exercise regimen experiments, including the frequency of monitoring and the criteria used to assess animal health and well-being.

Thank you for your attention to these requests.

4.  Please note that PLOS does not permit references to “data not shown.” Authors should provide the relevant data within the manuscript, the Supporting Information files, or in a public repository. If the data are not a core part of the research study being presented, we ask that authors remove any references to these data.

Reviewers' comments:

Reviewer's Responses to Questions

**Comments to the Author**

1. Is the manuscript technically sound, and do the data support the conclusions?

Reviewer #1: Partly

Reviewer #2: Partly

2. Has the statistical analysis been performed appropriately and rigorously? 

Reviewer #1: Yes

Reviewer #2: Yes

3. Have the authors made all data underlying the findings in their manuscript fully available?

Reviewer #1: Yes

Reviewer #2: Yes

4. Is the manuscript presented in an intelligible fashion and written in standard English?

Reviewer #1: No

Reviewer #2: Yes

5. Review Comments to the Author

Reviewer #1: The manuscript by Pillai et al, examines the effects of PCB exposure on wound healing and if exercise could alter the rate of would healing following PCB exposure. Although potentially interesting, the manuscript is written and data are presented in a confused way and the conclusion are not evidenced.

As it is, the manuscript is just a list of experiments without any link among them. And considering the lack of significant changes in wound healing rates in the presence of PCB exposure and fluctuations in the statistically significant differences in inflammatory markers at random time points, I am afraid there is not even a strong correlation of the outcomes and markers that have been measured. What we see here could just be the noise in the data (upregulation of certain markers that may not even related to minor would healing changes between the groups)

The authors should rewrite the manuscript following a more logic way for planned experiments. The authors should definitively rewrite the Discussion part that, as it is, this section reads more like results with little discussion. The authors should clearly state which is the main finding of the study and discuss it according to the literature.

Reviewer #2: Review of manuscript entitled “Exercise influences the impact of polychlorinated biphenyl exposure on immune function”, submitted for publication in Plos One.

The manuscript examines how PCB exposure at different doses (0, 100, 500, 1000 ppm i.p.) altered wound healing in exercised versus non-exercised subgroups of mice and investigates associated immune response modulation. Effect on wound size and alteration in key cytokines levels are measured following exposure of young female mice to exercise for two weeks followed by single dose PCB exposure. This study found that neither exercise nor PCB exposure significantly affect wound healing rate. Some significant alteration in cytokines are observed. The aim of the study was to study the mitigating effect of exercise against PCB effects on wound healing. Exercise had some varied effect on cytokines levels. I will suggest further consideration before manuscript is accepted.

1) The author should justify their choice of PCB doses.

2) The authors should reduce text and further deemphasize discussion of no statistically significant data. Also, some speculative language should be omitted (line 266-267).

3) Line 495-496 and elsewhere. it should be made clear that data are not statistically significant before discussing pattern in data.

4) I suggest data be presented as mean ± SD to better understand the said patterns in data.

5) It is not clear why cytokine levels are compared at days 3 and 5? It will be interesting to see a discussion on how cytokine changes found fit into the wound healing stages in mice.

6) It will be interesting to consider exploring sex differences in this study.

7) Explain significance symbols and related group comparison below each graph.

6. PLOS authors have the option to publish the peer review history of their article (what does this mean?). If published, this will include your full peer review and any attached files.

Reviewer #1: No

Reviewer #2: No

---

## [Author Response · Author response to Decision Letter 0]

26 Jun 2020

Response to reviewers has been attached as a separate file as well. Following is the same information from that response.

Hans-Joachim Lehmler, PhD

Academic Editor

PLOS ONE

Dear Dr. Lehmler

We set out to examine diverse factors of exercise, exposure to endocrine disruptors and immune responses to unravel possible crucial interactions. As with most of the work on endocrine disruptors as environmental toxins, the results express not a simple but more of a complex, intricate relation among variables, thereby opening up unique research possibilities and merging distinct fields to advance understanding. We sincerely thank you and the reviewers for the thoughtful comments and the opportunity to resubmit this experimental work, entitled “Exercise influences the impact of polychlorinated biphenyl exposure on immune function” to the journal, PLOS ONE.

 We have responded to your and reviewers comments inline (italicized, line numbers provided are based on the track changes version of the manuscript) below. This includes a substantial revision of the results and discussion sections. We have altered the presentation of the data emphasizing the significant results and producing a cohesive, ordered manner of presentation of the findings. The discussion provides an overview of related work, interprets the relevant data and does not overextend any one particular idea. A strength of the experiment and its findings includes the combination of different features involved and how the study deals with the interactions among these different factors. The reviewers mentioned other important factors not directly examined, and we now deliberate on these factors as they relate to the present work and portent future studies better designed to investigate novel ways to reduce the harmful impact of environmental toxin exposure.

 Please let me know if you have any additional questions or comments.

Best Wishes,

Dr. Mahesh Pillai

We have reviewed and met these requirements. The following sections were included in the initial submission, but since it was not found in the style template it was deleted.

FUNDING: Department of Biological Sciences, Bowling Green State University, Bowling Green, Ohio. The funders had no role in study design, data collection and analysis, decision to publish, or preparation of the manuscript.

COMPETING INTERESTS: The authors have no competing interests.

DATA AVAILABILITY STATEMENT: The data will be made available upon acceptance of the manuscript and that the repository information will be given upon submission of the manuscript.

2. We noted that one of your references did not auto populate and instead the manuscript contains the following "Citation", please replace this with the appropriate reference during your next revision.

 Citation added in line 66 

Please note: line numbers provided are based on the track changes version of the manuscript

3. At this time, we request that you please report additional details in your Methods section regarding animal care, as per our editorial guidelines:

(1) Please state the number of mice used in the study

 Table 1 (page 8) and Table 2 (page 9) show the detailed distribution of mice used in this study. 48 mice were used for the wound healing part and 96 mice were used for the cytokine analysis part of this study.

(2) Please include the secondary and confirmatory method of euthanasia, in addition to CO2 asphyxiation

Post CO2 asphyxiation, mice were decapitated with a guillotine (updated in line 162, 211, 226)

(3) Please provide the dosage of isoflurane used to anaesthetise the mice during the wound creation experiment

2% (updated in line 191)

(3) Please describe the post-operative care received by the animals in both the wound creation and exercise regimen experiments, including the frequency of monitoring and the criteria used to assess animal health and well-being.

After PCB exposure, exercise regimen and wound generation, the animals were frequently monitored for evidence of distress (poor grooming, frantic appearance, poor coat condition, aversion to handling and wound infection). No signs of distress were observed in animals in this study. (updated in line 160)

4. Please note that PLOS does not permit references to “data not shown.” Authors should provide the relevant data within the manuscript, the Supporting Information files, or in a public repository. If the data are not a core part of the research study being presented, we ask that authors remove any references to these data.

That reference has been removed from line 213

We have identified a repository for the data and will provide information about it as soon as the manuscript is accepted.

Reviewers' comments:

Reviewer's Responses to Questions

Comments to the Author

1. Is the manuscript technically sound, and do the data support the conclusions?

Reviewer #1: Partly

Reviewer #2: Partly

The data and the discussion have been significantly reworked on as per the reviewer’s suggestions.

Please note: line numbers provided are based on the track changes version of the manuscript

2. Has the statistical analysis been performed appropriately and rigorously? 

Reviewer #1: Yes

Reviewer #2: Yes

3. Have the authors made all data underlying the findings in their manuscript fully available?

Reviewer #1: Yes

Reviewer #2: Yes

4. Is the manuscript presented in an intelligible fashion and written in standard English?

Reviewer #1: No

Reviewer #2: Yes

 The data and the discussion have been significantly reworked on as per the reviewer’s suggestions.

5. Review Comments to the Author

Reviewer #1: The manuscript by Pillai et al, examines the effects of PCB exposure on wound healing and if exercise could alter the rate of would healing following PCB exposure. Although potentially interesting, the manuscript is written and data are presented in a confused way and the conclusion are not evidenced.

As it is, the manuscript is just a list of experiments without any link among them. And considering the lack of significant changes in wound healing rates in the presence of PCB exposure and fluctuations in the statistically significant differences in inflammatory markers at random time points, I am afraid there is not even a strong correlation of the outcomes and markers that have been measured. What we see here could just be the noise in the data (upregulation of certain markers that may not even related to minor would healing changes between the groups)

The authors should rewrite the manuscript following a more logic way for planned experiments. The authors should definitively rewrite the Discussion part that, as it is, this section reads more like results with little discussion. The authors should clearly state which is the main finding of the study and discuss it according to the literature.

The data and the discussion have been significantly reworked on as per the reviewer’s suggestions. The results have been re-arranged so that it provides better understanding of specific interactions: Exercise and Day, PCB and Day, PCB and Exercise. Effects of PCB have been ordered from 0 to 100 to 500 to 1000 in each section to improve readability. The discussion provides more details and comparisons to previous studies.

Reviewer #2: Review of manuscript entitled “Exercise influences the impact of polychlorinated biphenyl exposure on immune function”, submitted for publication in Plos One.

The manuscript examines how PCB exposure at different doses (0, 100, 500, 1000 ppm i.p.) altered wound healing in exercised versus non-exercised subgroups of mice and investigates associated immune response modulation. Effect on wound size and alteration in key cytokines levels are measured following exposure of young female mice to exercise for two weeks followed by single dose PCB exposure. This study found that neither exercise nor PCB exposure significantly affect wound healing rate. Some significant alteration in cytokines are observed. The aim of the study was to study the mitigating effect of exercise against PCB effects on wound healing. Exercise had some varied effect on cytokines levels. I will suggest further consideration before manuscript is accepted.

1) The author should justify their choice of PCB doses.

Explained in line 168. “Previous studies have used similar PCB dosage (Nishida et al., 1997) and intraperitoneal injections (Zhao et al., 1997).”

Please note: line numbers provided are based on the track changes version of the manuscript

2) The authors should reduce text and further deemphasize discussion of no statistically significant data. Also, some speculative language should be omitted (line 266-267).

This sentence has been reworded.

3) Line 495-496 and elsewhere. it should be made clear that data are not statistically significant before discussing pattern in data.

The data has been rearranged to look more cohesive and clear. This has been done by focusing on individual interactions between PCB, Day and exercise; and emphasizing significant data.

4) I suggest data be presented as mean ± SD to better understand the said patterns in data.

In the manuscript (Average ± SEM) is used and this is standard for other work.

5) It is not clear why cytokine levels are compared at days 3 and 5? It will be interesting to see a discussion on how cytokine changes found fit into the wound healing stages in mice.

Line 228 has explanation on why day 3 and 5 were used. “Previous study has measured cytokine levels in wound tissue at 3 and 5 days after wound creation (Keylock et al., 2008).” 2 paragraphs starting on line 1011 in discussion talk about cytokine changes in wound healing. We have discussed the role of cytokines in the inflammatory process and how the results from this study compare to previous studies.

6) It will be interesting to consider exploring sex differences in this study. 

2 paragraphs starting on line 965 in discussion talk about sex differences in this study and previous studies. We have discussed different effects of PCB exposure and exercise in relationship to sexual differences.

7) Explain significance symbols and related group comparison below each graph.

The authors have detailed the significance of symbols and related group comparisons in the results section and believe that it will be redundant to include this information below each group. The information is present is sections describing results from each figure.

---

## [Decision Letter · Decision Letter 1]

3 Aug 2020

Exercise influences the impact of polychlorinated biphenyl exposure on immune function

PONE-D-20-09774R1

Dear Dr. Meserve,

We’re pleased to inform you that your manuscript has been judged scientifically suitable for publication and will be formally accepted for publication once it meets all outstanding technical requirements.

Kind regards,

Hans-Joachim Lehmler, PhD

Academic Editor

PLOS ONE

Additional Editor Comments (optional):

The decision to accept this manuscript is based on the feedback by Reviewer #1 and my own evaluation of the manuscript. 

Reviewers' comments:

Reviewer's Responses to Questions

**Comments to the Author**

1. If the authors have adequately addressed your comments raised in a previous round of review and you feel that this manuscript is now acceptable for publication, you may indicate that here to bypass the “Comments to the Author” section, enter your conflict of interest statement in the “Confidential to Editor” section, and submit your "Accept" recommendation.

Reviewer #1: All comments have been addressed

Reviewer #2: (No Response)

2. Is the manuscript technically sound, and do the data support the conclusions?

Reviewer #1: Yes

Reviewer #2: (No Response)

3. Has the statistical analysis been performed appropriately and rigorously? 

Reviewer #1: Yes

Reviewer #2: Yes

4. Have the authors made all data underlying the findings in their manuscript fully available?

Reviewer #1: Yes

Reviewer #2: Yes

5. Is the manuscript presented in an intelligible fashion and written in standard English?

Reviewer #1: Yes

Reviewer #2: No

6. Review Comments to the Author

Reviewer #1: (No Response)

Reviewer #2: The data presented are not strong enough to support the conclusion. The significant effects observed in inflammatory response appears random. The association between wound healing and measured cytokines relied at based on trends in data with great variations in data points.

7. PLOS authors have the option to publish the peer review history of their article (what does this mean?). If published, this will include your full peer review and any attached files.

Reviewer #1: No

Reviewer #2: No

---

## [Editor Report · Acceptance letter]

12 Aug 2020

PONE-D-20-09774R1 

Exercise influences the impact of polychlorinated biphenyl exposure on immune function 

Dear Dr. Meserve:

I'm pleased to inform you that your manuscript has been deemed suitable for publication in PLOS ONE. Congratulations! Your manuscript is now with our production department. 

Kind regards, 

on behalf of

Dr. Hans-Joachim Lehmler 

Academic Editor

PLOS ONE